# Enhancing PD-1 Gene Silence in T Lymphocytes by Comparing the Delivery Performance of Two Inorganic Nanoparticle Platforms

**DOI:** 10.3390/nano9020159

**Published:** 2019-01-28

**Authors:** Yanheng Wu, Wenyi Gu, Li Li, Chen Chen, Zhi Ping Xu

**Affiliations:** 1Australian Institute for Bioengineering and Nanotechnology, the University of Queensland, St. Lucia 4072, QLD, Australia; yanheng.wu@uq.net.au (Y.W.); l.li2@uq.edu.au (L.L.); 2School of Biomedical Sciences, the University of Queensland, St. Lucia 4072, QLD, Australia; chen.chen@uq.edu.au

**Keywords:** layered double hydroxide (LDH) nanoparticle, lipid-coated calcium phosphate (LCP) nanoparticle, programmed cell death protien-1 (PD-1), human tumor infiltrating lymphocytes (TILs), PD-1 gene silencing, EL4 cells

## Abstract

Suitable carriers are crucial to RNAi applications for cancer genotherapy and T-cell immunotherapy. In this research, we selected two extensively-investigated biocompatible inorganic nanoparticle carriers, i.e., layered double hydroxide (LDH) and lipid-coated calcium phosphate (LCP) and then compared their efficacy for siRNA delivery in T cells, in order to understand which carrier is more efficient in delivering functional programmed cell death protein 1 siRNA (PD-1 siRNA) to suspended T lymphocytes. Both LDH and LCP nanoparticles quickly delivered gene segment to mouse T cell lines (EL4), while the LCP nanoparticles exhibited more cellular uptake and higher PD-1 gene silence efficiency. We further demonstrated that LCP nanoparticles successfully reduced the expression of PD-1 in human ex vivo tumor infiltrating lymphocytes (TILs). Thus, LCP nanoparticles can be used as a better nano-carrier for gene therapy in lymphocytes, especially in regards to TIL-related cancer immunotherapy.

## 1. Introduction

RNAi technology has been now examined for enhanced cancer immunotherapy in combination with other therapies [1,2,3,4]. For example, silencing programmed death ligand 1 (PD-L1) on the surface of cancer cells is able to induce higher T-cell immunity and enhance the therapeutic efficacy in combination with photodynamic therapy [4]. To this end, many research inquiries have been conducted to knockdown PD-L1 expression on tumor cells, while silencing programmed death 1 (PD-1) on the surface of T cells is rarely investigated. An efficient siRNA delivery vehicle is necessary to knockdown the expression of PD-L1 on tumor cells or PD-1 on the T cells. For gene delivery, many types of nanoparticles (NPs), such as polymeric NPs (PEI), biomolecular NPs (PLGA and BSA), and inorganic NPs (Au, carbon, and SiO_2_) are extensively investigated [5], while two inorganic nanoparticles (NPs), i.e., layered double hydroxide (LDH) [6,7,8,9,10] and lipid-coated calcium phosphate (LCP) [11,12,13,14,15], both with many suitable properties, appear as efficient delivery vehicles for the functional siRNA.

LDH is a group of anionic clay materials that have attracted increasing attention in recent years for biomedical applications, such as gene delivery, vaccine delivery, and drug delivery [6,7,8,9,10,16,17]. In particular, MgAl-LDH nanoparticles (NPs) have been demonstrated as efficient vehicles for the delivery of genes and drugs to cells [6,7,8,9,10,16], which are biocompatible, have a high loading capacity, and release target biomolecules in a pH-dependent manner [17,18]. More advantages of LDH NPs as delivery vehicles include the low toxicity, protection of payloads, and their high cellular delivery efficacy [7,10,17,19]. These properties enable LDH NPs to become a good option for the cellular delivery of DNAs or RNAs. Furthermore, to overcome poor colloidal stability due to aggregation in biological media for in vivo applications, we recently developed an approach to coating bovine serum albumin (BSA) on LDH nanoparticles (BSA-LDH), which colloidally stabilizes the LDH NPs in various electrolyte solutions and media [20,21].

On the other hand, calcium phosphate (CaP), which is the main inorganic component in bones, has excellent properties as a nanocarrier of DNA and siRNA for gene therapy in the nanomaterial form, as reported elsewhere in the last decades [5,22]. Recently, by stabilizing the CaP NPs with a lipid bilayer, Huang et al. developed LCP NPs that naturally possess a colloidal stability in electrolyte solutions and biological media. LCP NPs have been demonstrated to significantly improve siRNA delivery when compared with their lipid/protamine/DNA (LPD) formulation [13,23]. We have recently shown that LCP NPs improved the cellular uptake of siRNA and they significantly inhibited the growth of human breast cancer cells in vitro with our optimized LCP NPs [14,15,24].

Both LDH and LCP NPs appear as good siRNA delivery systems. However, which one is more efficient has not been investigated and reported yet. Therefore, in this study, we compared the capability of LDH and LCP NPs in delivering murine PD-1 siRNA into EL4 T cells. EL4 T cells are cancerous lymphocytes, with high PD-1 expression, which are a suitable model for comparing the delivery systems in terms of the silence efficacy of target gene PD-1. Thus, the objectives of this research were to: (1) reveal the time-dependent and dosage-dependent cellular uptake of LDH NPs, BSA-LDH NPs, and LCP NPs by EL4 cells; (2) understand the factors that affect the siRNA delivery efficacy in terms of target PD-1 mRNA silence and protein expression reduction in EL4; and, (3) confirm that the selected better system works on a human T cell, i.e., tumor infiltrating lymphocyte (TIL). Here, TILs were isolated from breast cancer patients with high PD-1 expression and then transfected with the selected system to confirm the efficient knockdown of PD-1 gene expression. Our findings in this research suggest a set of optimal parameters and a better delivery system for PD-1 silence on TILs as well as other cancer cells and lymphocytes.

## 2. Materials and Methods

### 2.1. Chemicals and Reagents

All of the samples were prepared under sterile conditions. Sodium hydroxide pellets, magnesium chloride hexahydrate (MgCl_2_·6H_2_O), and aluminum chloride hexahydrated (AlCl_3_·6H_2_O) were purchased from Ajax Finechem (Taren Point, NSW, Australia), and Sigma-Aldrich (Castle Hill, NSW, Australia), respectively. dsDNA-Cy5 was purchased from GeneWorks (San Diego, CA, USA). PD-1 siRNA (sense: 5′-AGAcGuAAGcAGuGuuGAAdTsdT-3′ and antisense: 5′-UUcAAcACUGCUuACGUCUdTsdT-3′ for EL4, and sense 5′-AGAccuuGAuAcuuucAAAd-TsdT-3′ and antisense 5′-UUUGAAAGuAUcAAGGUCUdTsdT-3′ for human TILs), and other chemicals and reagents were purchased from Sigma-Aldrich (Castle Hill, NSW, Australia) if not illustrated specifically. Water used in experiments was deionized Milli-Q water.

### 2.2. LDH NP Preparation

Mg_2_Al-Cl-LDH NPs were synthesized by the co-precipitation-hydrothermal method, which had been well established in our group [25,26]. Briefly, a mixture of MgCl_2_ (0.70 M) and AlCl_3_ (0.30 M) with a total volume of 10 mL was quickly added to 40 mL of NaOH solution (0.45 M) within 5 s, under vigorous stirring. After 10 min stirring, the slurry was separated via centrifugation and then re-dispersed in 40 mL of deionized water. The resultant suspension was moved into a stainless steel autoclave with a Teflon lining and then heated at 100 °C for 16 h. The final mass concentration of LDH was approximately 10 mg/mL, with the yield of ~60%.

BSA were added to stabilize LDH particles in medium, as reported previously [21]. Two milliliters of 4 mg/mL LDH suspension was added into 2 mL of 10 mg/mL BSA solution (dissolved in deionized Milli-Q water) drop by drop under vigorously magnetic stirring for 30 min to ensure saturated absorption.

Specifically, Cy5-dsDNA/siRNA was loaded directly onto LDH NPs by adding to the LDH suspension at the LDH/dsDNA or siRNA mass ratio of 10:1, which was then diluted to the designed volume for cellular uptake or cell transfection after 15 min incubation at room temperature.

### 2.3. LCP NPs Preparation

LCP NPs were prepared by a modified two-step method that was based on the previous reports [13,24]. The anionic lipid coated calcium phosphate (CaP) core was synthesized by a water-in-oil microemulsion method and then the CaP core was coated with a second lipid layer to form the bilayer lipid-coated CaP (LCP) nanoparticles (NPs) by the film-rehydration method. Briefly, the first microemulsion (solution 1a) was prepared by mixing 75 µL of 5 M CaCl_2_ and 50 µL of H_2_O with 5 mL of premixed cyclohexane/Igepal CO-520 (70/30, *v/v*). The second microemulsion (solution 1b) containing sodium phosphate was prepared by adding 75 µL of 50 mM Na_2_HPO_4_ and 50 µL of H_2_O into another 5 mL of oil phase. Subsequently, the second microemulsion was added with 50 µL of 20 mM DOPA in chloroform and then with the first microemulsion drop by drop, which was followed by stirring for another 15 min. The CaP-DOPA cores were harvested by adding 10 mL of absolute ethanol and centrifuging at 10,000× g for 20 min, and then being washed with 10 mL ethanol three times. The collected CaP core particles were then dispersed in 1 mL of CHCl_3_ and mixed with DOTAP/DOPC/cholesterol (2:1:3). After evaporation under reduced pressure, the lipid film was then hydrated in PBS buffer (pH = 7.4) to obtain LCP NPs, which were normally well dispersed under gentle ultrasound treatment. Similarly, Cy5-dsDNA or siRNA was loaded in LCP NPs by adding the same amount of Cy3-dsDNA or siRNA (15 µL, 100 µM) in both CaCl_2_ and Na_2_HPO_4_ solutions (75 µL), followed by the same preparation procedures. The dsDNA/siRNA loading capacity was approximately 58 µg/mg (LCP), as reported previously [24].

### 2.4. Characterization of LDH, BSA-LDH and LCP NPs

The particle size distribution and zeta potential of as-prepared LDH, BSA-LDH, and LCP NPs was determined by photon correlation spectroscopy (PCS, Zetasizer Nano ZS, MALVERN Instruments, Malvern, UK) using Non-Invasive Backscatter optics (NIBS). For Transmission electron microscope (TEM) imaging, LDH, BSA-LDH, and LCP solution was air-dried on a copper grid. The images were obtained on a JEOL 1010A transmission electron microscope (Tokyo, Japan) at an acceleration voltage of 200 kV.

### 2.5. Cell Culture and FACS Analysis

EL4 cells, which constitutively produce PD-1, a widely studied Mouse lymphoma cell line, was used as the cancer cell model in this study. Normally, EL4 cells were cultured in DMEM that was supplemented by 10% (*v/v*) FBS at 37 °C in 5% CO_2_ atmosphere.

As-cultured EL4 cells were collected by centrifugation. After three washes with 2% FCS/PBS, the cells were fixed in 2% paraformaldehyde/PBS. For FACS analysis of PD-1 expression, the cells were then stained with PD-1 antibody (Cell Signal Technology, 1:500 dilution), and then analyzed in a BD Accuri™ C6 (San Jose, CA, USA) flow cytometer with CFlow Sampler software (Becton Dickinson, Mountain View, CA, USA).

### 2.6. Tumor infiltrating lymphocyte Isolation and Characterization

Tumor infiltrating lymphocytes (TILs) were isolated from breast cancer biopsy specimens by mincing the tissue into small pieces and then digesting them with collagenase type IV (0.1 mg/mL) (Sigma-Aldrich, Castle Hill, NSW, Australia) for 2 h, followed by culture in X-VIVO-15 medium (Lonza, Basel, Switzerland) containing 5% human AB serum and recombinant human IL-2 (150 IU/mL) in 24-well plates, followed by an expansion using a rapid expansion protocol (REP) [27,28]. Once a sufficient number of T cells (>1 × 10^7^) was generated, they were cryopreserved for later expansion. A REP for “young TILs” that was previously used in melanoma patients was followed. Cryopreserved, pre-REP TILs from patients were thawed and then further expanded to treatment levels using an anti-CD3 antibody (OKT-3, 30 ng/mL, R&D Systems, Minneapolis, MN, USA), rhIL-2 (BD PharMingen, San Jose, CA, USA), and irradiated feeder cells, as previously described [28]. The expanded TILs were fixed and then labelled with human CD4 and CD8 antibody conjugates, followed by labelling human PD-1 antibody conjugate and being subjected to FACS analysis.

### 2.7. Delivery of dsDNA and siRNA to EL4 and TILs

About 1 × 10^6^ cells EL4 cells or TILs were seeded in the wells of six-well plates and then mixed with LDH-PD-1-siRNA/Cy5-dsDNA or LCP-PD-1-siRNA/Cy5-dsDNA at 37 °C for a period of time (0–8 h) at 20–80 nM of siRNA or dsDNA. The culture medium was then replaced with the fresh medium and transfected EL4 or TILs were cultured for another 72 h at 37 °C, 5% CO_2_. After transfection, the cells were collected and the PD-1 expression level was determined by RT-PCR, ELISA assay, and FACS, respectively.

### 2.8. Western Blotting

The relative PD-1 concentration in cell lysates was estimated using the Pierce™ BCA Protein Assay Kit (Thermo Fisher Scientific, Waltham, MA, USA). The samples (roughly 10 µg total protein per well) were mixed with protein-loading buffer (Bio-Rad, Hercules, CA, USA) containing 2-mercaptoethanol. After denaturation by 5-min boiling, the samples were loaded on 4–15% Mini-PROTEAN^®^ TGX™ Precast Gels SDS polyacrylamide gel (Bio-Rad). Gels were blotted onto polyvinylidene difluoride (PVDF) membranes for 90 min at 80 V, the proteins were transferred onto PVDF membranes and then blocked for non-specific binding with TBST (0.05% (*v/v*) Tween-20 in TBS pH 7.4) plus 5% BSA for 1 h at RT. PD-1 RabMab antibodies (ab205921) (Abcam, Cambridge, UK; at dilution of 1:800–1000) was applied overnight at 4 °C. The membrane was washed with TBST and then incubated with HRP-conjugated secondary antibodies (Goat anti-rabbit IgG H&L (HRP), ab97051) (Abcam, Cambridge, UK; at dilution 1:5000) for 2 h. After washing, the protein bands were visualized using Clarity™ Western ECL Substrate (Bio-Rad) and then analyzed using ImageJ v1.40 software (Hercules, CA, USA).

### 2.9. Real-Time PCR

All real-time PCR assays were conducted according to the manufacturer’s instructions. In brief, 1.2 mL of trizol/chloroform (1:5, *v/v*) was added to lyse cells and the supernatant was then collected by centrifugation (12,500 rpm, 15 min). Subsequently, 2.4 mL of isopropanol was added and the supernatant was centrifuged for another 15 min at 12,500 rpm. The collected pellet was washed with 70% ethanol. After drying the pellet, 50 µL of H_2_O was added to resuspend the total RNA. Reverse transcription reactions were performed in 20 µL as the manufacturer instructed and the cDNA was diluted with 80 µL of H_2_O. For each well, 3.5 µL of cDNA solution that was mixed with 8.5 µL of PCR Master-Mixture. After centrifugation, real time PCR was conducted (Real Time PCR, iCycler iQ^TM^, Bio-Rad, Hercules, CA, USA).

### 2.10. Statistical Analysis

Data are presented as the mean ± SEM or the mean ± SE and analyzed by two-way ANOVA using GraphPad Prism software. All of the cell tests were done in triplicate. A *p*-value < 0.05 was considered to be significant. * *p* < 0.05; ** *p* < 0.01; *** *p* < 0.001.

## 3. Results

### 3.1. Physicochemical Properties of LDH and LCP NPs

Homogeneously dispersed Mg_2_Al-Cl-LDH NPs had a narrow particle size distribution (Figure 1A,D), with the equivalent mean hydrodynamic diameter of 110 nm and the polydispersity index (PDI) of 0.099. The LDH NP suspension was transparent, with a zeta potential of 35 mV. The TEM image (Figure 1A) shows that LDH crystallites were well crystallized, with a typical hexagonally shaped morphology. These observations are consistent with previous reports [25,26]. As also shown in Figure 1B,E, bovine serum albumin (BSA) coated LDH (BSA-LDH with the BSA/LDH mass ratio of 5:2 and the LDH concentration of 2.0 mg/mL) had an average size of 176 nm with a PDI of 0.229 and a zeta potential of about −20 mV. These data mean that LDH NPs were well coated with BSA and then colloidablly stabilized with BSA, which is consistent with previous reports [20,21]. The TEM image in Figure 1B clearly shows that BSA-LDH (5:2) nanocomplexes were nearly mono-dispersed in PBS. The size increase suggests that LDH-BSA NPs were slightly aggregated via the BSA bridging effect [20].

The average size of LCP NPs was about 40 nm after the CaP cores were coated with the lipid bilayer (Figure 1C,F), with a zeta potential of 18 mV [24]. The LCP NPs were well dispersed and sphere-like particles, as observed by TEM (Figure 1C). The TEM image confirmed the typical structure of LCPs, containing a CaP core and a coating lipid membrane [23,24]. When the dsDNA or siRNA was loaded (58 µg/mg CaP), the average particle size was similar, while the zeta potential was reduced to about 14 mV [24].

These data indicate that the LDH, BSA-LDH, and LCP nanoparticles that were used in this study possess the typical physicochemical properties of LDH and LCP NPs, as reported previously [20,24].

### 3.2. Cellular Uptake of LDH, BSA-LDH and LCP NPs

Cy5-dsDNA was bound to LDH or encapsulated within LCP NPs and used as the dye tag to quantify the cellular uptake. As shown in Figure 2A, different uptake behaviors by EL4 were observed. For LDH and BSA-LDH, the peak of uptake percentage was achieved at 2–4 h (~50% and ~40%, respectively), with the uptake amount then being slightly decreased. Although EL4 cells were suspended, more LDH-dsDNA was taken up by EL4 than LDH-BSA-dsDNA, which is probably due to the positive zeta potential of LDH-dsDNA (10:1, 20–30 mV) [29], in comparison with that of LDH-BSA-dsDNA (more negative than −20 mV after dsDNA was loaded onto LDH-BSA). The amount decrease after 4 h might be due to the metabolization of Cy5-dsDNA released into the cytosol after endocytosis of LDH-Cy5-DNA.

For LCP NPs, the cellular uptake reached the saturation at 4 h (68%) and was maintained for another 4 h. When compared with LDH and BSA-LDH, LCP NPs have a 20–30% higher uptake amount. This increase is largely attributed to the monodispersed stability of LCPs in medium and the positive surface charges, which provides more chances for the NP attachment to the negatively-charged membrane of suspended EL4 cells, leading to a higher cellular uptake.

As shown in Figure 2B, LCP NPs also showed a consistently higher cellular uptake at three Cy5-dsDNA doses (20, 40, and 80 nM). The positive cell percentage in the BSA-LDH NPs treated group was 20%, 55%, and 71%, which was then increased to 24%, 61%, and 79% for the LDH NPs treated group, and 35%, 75%, and 91% for the LCP NPs treated group, respectively. Based on these data, we therefore conclude that LCP NPs are more effective in delivering dsDNA into EL4 cells.

### 3.3. PD-1 Expression and Gene Silence in EL4 Cells

To examine the delivery efficacy of LCP and LDH NPs for gene silence in EL4, we firstly analyzed PD-1 expression level in EL4 cells. EL4 cells were labelled with mouse CD4 and CD8 antibody conjugates, followed by labelling murine PD-1 antibody conjugate. The FACS data shows that 57.4% CD4+ EL4 cells and 32.0% CD8+ EL4 cells were PD-1 positive cells (Figure 3A). Overall, EL4 cells had a relatively high percentage of cells that were PD-1 positive.

EL4 cells were then treated with LDH-, BSA-LDH- and LCP-siRNA-PD-1 at 80 nM of siRNA-PD-1. As displayed in Figure 3B, the PD-1 mRNA expression in EL4 was decreased by 13% and 29% using BSA-LDH-siRNA-PD-1 and LDH-siRNA-PD-1, respectively. In sharp contrast, 72% PD-1 mRNA expression was reduced in EL4 cells using LCP-siRNA-PD-1. Consistently, the PD-1 protein expression level was downregulated to 80%, 78%, and 22% while using BSA-LDH-siRNA-PD-1, LDH-siRNA-PD-1, and LCP-siRNA-PD-1 nanohybrids (Figure 3C), respectively. The PD-1 gene silence, both in terms of mRNA and protein expressions, is actually consistent with the cellular uptake of these LDH and LCP NPs by EL4 cells (Figure 2), which all suggest that LCP NPs are a more effective delivery system for EL4 cell uptake and gene silence. Note that the scramble siRNA that was delivered by LCP did not cause the knockdown of the target PD-1 gene (data not shown).

### 3.4. PD-1 Expression and Gene Silence in Human TILs

Human TILs from breast cancer patients were isolated and then expanded in vitro. The cells were labelled with anti-human CD4 and CD8 antibody conjugates and PD-1 expression was also labelled with a specific fluorescence antibody. The CD4+ and CD8+ cells were gated for PD-1 expression. The FACS data shows that CD4 and CD8 positive cells had a similar PD-1 positive cell percentage (71.3% vs. 68.0%) (Figure 4).

The expanded TILs were then treated with LCP-siRNA-PD-1 NPs to downregulate the expression of PD-1 mRNA and the PD-1 protein. As shown in Figure 5A, the PD-1 mRNA percentage was significantly decreased from 73% to 33% when the siRNA-PD-1 concentration was increased from 20 to 80 nM. As shown in Figure 5B, the PD-1 protein expression level was obviously reduced at the siRNA-PD-1 of 40 nM (by 32%), and even more at 80 nM (by 62%). The efficient silence of PD-1 gene can be largely attributed to the high delivery efficacy of LCP NPs for siRNA to T cells, as discussed previously [24].

TILs that were treated with LCP-siRNA-PD-1 at 80 nM of siRNA-PD-1 also showed a significant reduction of the PD-1 protein expressed on the TIL cell surface. As displayed in Figure 5C, the peak shifted from 2480 to 960 in terms of the mean florescent intensity, in consistence with 60% reduction of the PD-1 protein expression under similar conditions (Figure 5B). All of the data indicate that siRNA delivered by LCP NPs in this study can effectively downregulate PD-1 expression by T cells, such as EL4 and TILs.

## 4. Discussion

RNAi therapy always requires the efficient delivery to target cells. As demonstrated previously, both LDH and LCP NPs have high biocompatibility, good biodegradability, and low toxicity. As reported, the cell viability is >90% at the concentration of >200 µg/mL [24,30]. Thus, both can be effective and safe siRNA carriers for PD-1/PD-L1-based immunotherapy. In this work, both LDH and LCP NPs were further demonstrated to facilitate the delivery of dsDNA and siRNA to EL4 cells (Figure 2), cancerous lymphocytes that are homogeneously suspended in culture medium. The cellular uptake of various nanoparticles usually undergoes the clathrin-mediated endocytosis. Different from polymeric (such as polyethyleneimide) and biomolecular (such as BSA) NPs, which escape from the lysosomal pathway during endocytosis [31], LDH and LCP NPs escape from the endosomal pathway [9,12,17,32]. This is due to the anti-acidification of LDH (simplified as Mg_2_Al(OH)_6_Cl·1.5H_2_O, MW 240) and LCP (CaHPO_4_·0.5H_2_O, MW 145) materials when the endosome is acidified to later endosome, but before it is fused with the lysosome, because both NPs may dissolve within the endosome in the following ways:Mg_2_Al(OH)_6_Cl·1.5H_2_O + 6H^+^ → 2Mg^2+^ + Al^3+^ + Cl^−^ + 7.5H_2_O(1)
CaHPO_4_·0.5H_2_O + H^+^ → Ca^2+^ + H_2_PO_4_^−^(2)

The dissolution of LDH and LCP NPs leads to the increase of dissolved salt concentration, which causes the influx of water into the endosome due to the osmotic pressure. The continuous influx of water increases the volume and finally bursts the endosome, releasing NPs into the cytosol. As reflected in Figure 2, both LDH-Cy5-dsDNA and LCP-Cy5-dsDNA NPs were quickly taken up by EL4 cells in the first 2–3 h, and the maximum uptake was achieved at around 3 h.

Relatively, the LCP NPs delivered more Cy5-dsDNA into EL4 cells than LDH NPs (70% vs. 40–50% positive cell population, Figure 2), which could be attributed to their smaller size (40 vs. 110–170 nm, Figure 1). As reported elsewhere, cells take up NPs with the size of 50 nm much more quickly than that of 100–200 nm [16,33]. Moreover, the anti-acidification in Equation (A2) generates stronger ionic strength than in Equation (A1) per unit H^+^ number (e.g., 1 mmol) (Appendix A), which leads to more water influx and thus quicker escape from the endosome. The endosome escape is superior to the lysosome escape in protection and sustainable release of the payload. In the lysosomal pathway, most of the NP-siRNA hybrids are digested in lysosome, and a considerable amount of siRNA is destroyed, while in the endosome pathway, most of the siRNAs are still associated with NPs and then protected when they escape from the endosome. These associated siRNAs can be sustainably released in the cytosol for the continuous silencing of target gene, while there is less siRNA available for the release from polymer-siRNA hybrids after the lysosomal escape.

More interestingly, LCP-siRNA NPs more efficiently silenced the PD-1 gene expression in EL4 cells, in comparison with LDH NPs (Figure 3). Apart from the quick uptake by EL4 cells, there may be two other reasons. The main reason is that a unit H^+^ number releases more siRNA in the LCP NP form than in the LDH NP form. As explained in the Appendix A, 1 mmol of H^+^ is able to dissolve 145 mg of LCP and thus release 8.4 mg of siRNA for silence, while this amount of H^+^ can only dissolve 40 mg of LDH and thus release 4.0 mg of siRNA in the current experimental conditions. This estimation suggests that the functional siRNA released from LCP-siRNA hybrids is two times as that from LDH-siRNA hybrids during the endosome escape. On the other hand, the solubility of CaHPO4 is ca. 200 mg/L, while that of LDH is 50–100 mg/L [16] at the neutral pH, so more functional siRNA can be released from LCP-siRNA hybrids (11.6 mg/L siRNA) than from LDH-siRNA hybrids (5–10 mg/L siNRA) after they are delivered to the cytosol with the nearly neutral pH. Thus, these two factors, together with the quicker cellular uptake, enable LCP-siNRA hybrids to more efficiently release functional siRNA in the cytosol and knockdown the target PD-1 gene expression (Figure 3), seemingly being 3–4 times higher at the siRNA dose of 80 nM. When compared with optimized functional LDH-siRNA formulations (LDH:siRNA mass ratio = 5:1) reported previously [34], the current LCP NPs appear as more cost-effective siRNA carriers and vehicles for cellular delivery. As an example for confirmation, LCP NPs were further used to deliver the functional siRNA to human TILs and effectively knockdown the PD-1 gene expression (Figure 5).

In summary, the current research has demonstrated that LCP NPs are more efficient siRNA delivery vehicles than LDH in terms of cellular delivery efficacy and the knockdown efficiency of target gene expression.

## 5. Conclusions

In conclusion, we compared two inorganic nanoparticles, i.e layered double hydroxide (LDH) and lipid-coated calcium phosphate (LCP), which are both safe and effective vectors for siRNA delivery in many cell types regarding the delivery efficacy of functional siRNA (PD-1 siRNA) into suspended T lymphocytes (EL4) and silencing the target gene. We found that LCP NPs showed a more cellular uptake and higher PD-1 gene silence efficiency in mouse T cell line EL4 than LDH and BSA-LDH NPs. We further found that LCP NPs significantly reduced the expression of PD-1 in human ex vivo TILs, indicating the feasibility of using LCP NPs for gene therapy in lymphocytes, especially for TIL-related cancer immunotherapy.

## Figures and Tables

**Figure 1 nanomaterials-09-00159-f001:**
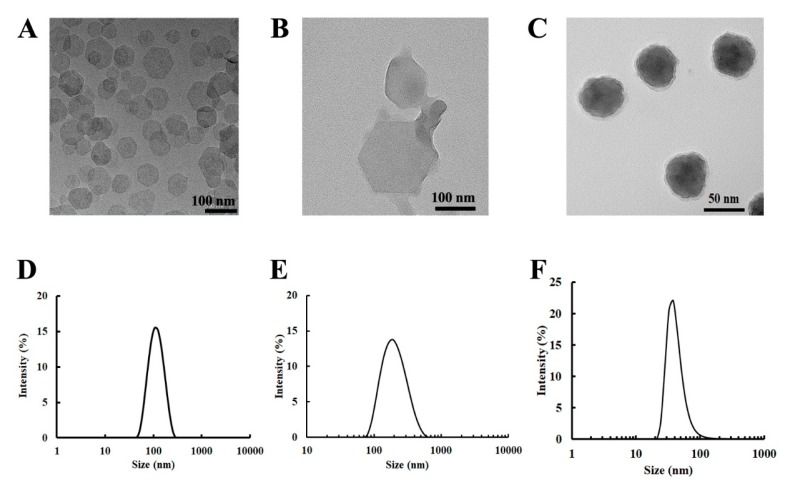
Particle morphology and particle size distribution. The upper penal shows the morphology of the nanoparticles used in this study (**A**) Mg_2_Al-Cl-LDH; (**B**) bovine serum albumin (BSA) on layered double hydroxide nanoparticles (BSA-LDH) in PBS; and, (**C**) lipid-coated calcium phosphate (LCP). The lower panel shows the size distributions of the corresponding nanoparticles (NPs) in the upper panel (i.e., (**A**,**D**); (**B**,**E**); and, (**C**,**F**)).

**Figure 2 nanomaterials-09-00159-f002:**
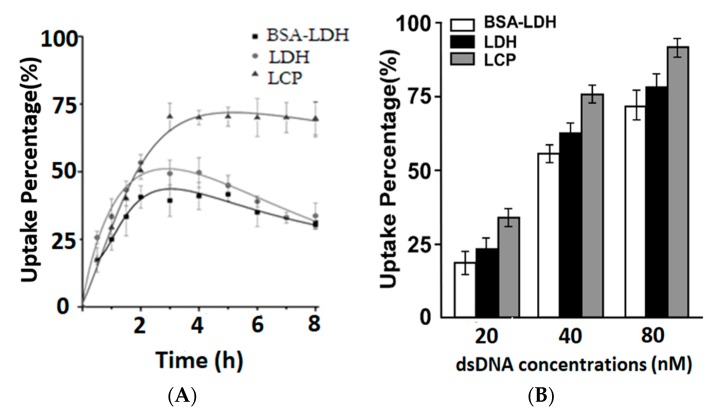
Cellular uptake profile of LDH and LCP NPs. (**A**) The fluorescence-activated cell sorting (FACS) results show the positive cell percentage vs. the time course (culture time) for EL4 cells treated with LDH-dsDNA, BSA-LDH-dsDNA and LCP-dsDNA hybrids at 40 nM of Cy-dsDNA. (**B**) The FACS results show the positive cell percentage vs. the treatment dose for EL4 cells treated with LDH-dsDNA, BSA-LDH-dsDNA and LCP-dsDNA hybrids for 4 h. All of the tests were done in triplicate.

**Figure 3 nanomaterials-09-00159-f003:**
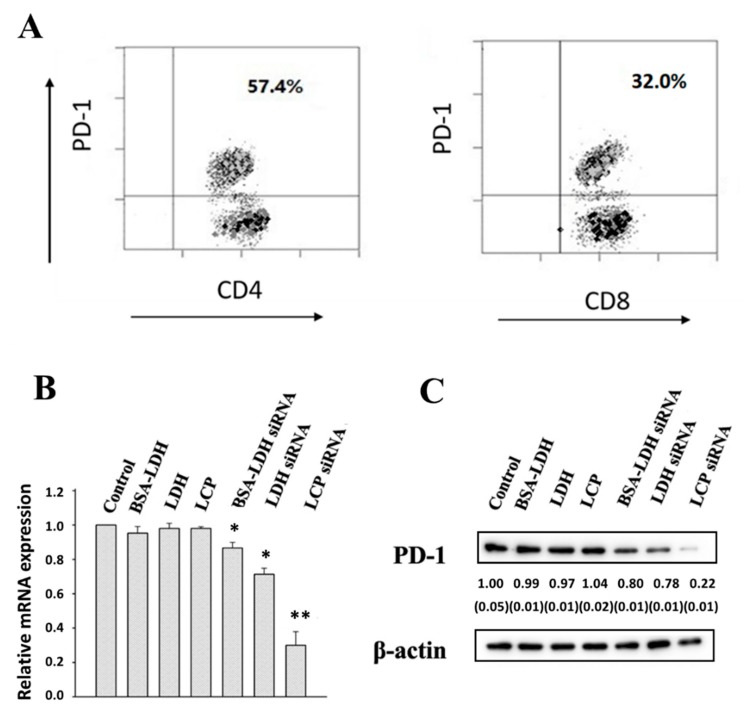
PD-1 expression and down-regulation of PD-1 in EL4. (**A**) PD-1 expression in gated CD4 and CD8 positive T cells (32.0%). (**B**) Real-time PCR data for the knockdown efficiency of PD-1 mRNA in EL4 treated with BSA-LDH-siRNA-PD-1; LDH-siRNA-PD-1; LCP-siRNA-PD-1 at the concentration of 80 nM for 4 h; and, (**C**) Western-blot showing the reduction of PD-1 expression in EL4 cells treated similarly. All tests were done in triplicate, with the SEM being listed in the parenthesis.

**Figure 4 nanomaterials-09-00159-f004:**
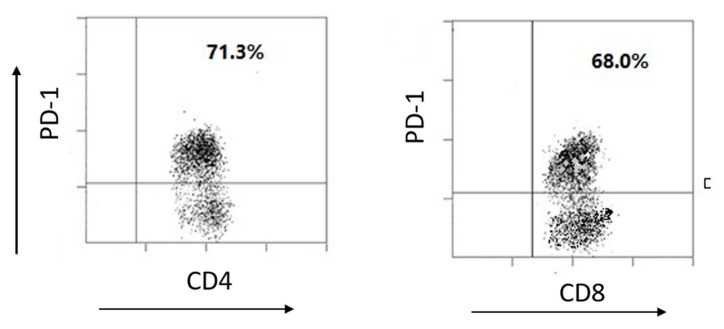
PD-1 expression in human tumor infiltrating lymphocytes (TILs). FACS results show the PD-1 positive cells in CD4 and CD8 cell populations from TILs.

**Figure 5 nanomaterials-09-00159-f005:**
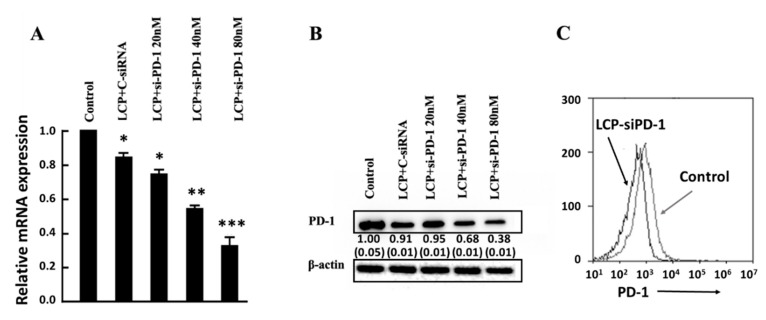
Down-regulation of PD-1 in TILs using LCP NPs. (**A**) Real-time PCR data for the knockdown efficiency of PD-1 mRNA in TILs treated with LCP+si-PD-1 at the concentration from 20 to 80 nM for 4 h; (**B**) Western-blot showing reduction of PD-1 expression in TILs treated similarly. The tests were done in triplicate, with the SEM listed in the parenthesis. (**C**) Flow cytometry of TILs before and after treatment with LCP-si-PD-1 at 40 nM. All of the tests were done in triplicate.

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
