# Peer review of "Enhancing PD-1 Gene Silence in T Lymphocytes by Comparing the Delivery Performance of Two Inorganic Nanoparticle Platforms"

_nanomaterials, 2019, doi:10.3390/nano9020159_

Reviewer 1 Report

FIGURE 2 B. X-axis labelling needs improvement.

FIGURE 5B  and elsewhere PD-1 protein is analyzed, a quantitative protein level graph should be included along with SEM

Author Response

Thanks for the reviewer's comments. The following lists the point-to-point responese.

1. FIGURE 2 B. X-axis labelling needs improvement. 

Responses: Figure 2A and 2B were redited and improved.

2. FIGURE 5B  and elsewhere PD-1 protein is analyzed, a quantitative protein level graph should be included along with SEM.

Responses:  The quantitative protein level (Figure 3C and 5B) was shown with the value of the band intensity in relation to the control one, with the SEM being listed in the parenthesis. As the information is already presented, so the bar graph may be not necessary in this situation. 

In addition the authors did many minor revisions to improve the language and results, as highlighted in red font.

Reviewer 2 Report

Wu et al. compared the performance of PD-1 siRNA delivery to T lymphocytes of layered double hydroxide and lipid-coated calcium phosphate carriers. They used mouse EL4 cells and isolated tumor infiltrating lymphocytes from human breast cancer as models. Both carriers were able to deliver the siRNA to lymphocytes but the LCP particles were more efficient. These carriers have been used by the group successfully already for inhibition of growth of human breast cancer cells.

Comments:

- Although it is known that the carriers are biocompatible it would be useful to know at which concentration they become cytotoxic in these cells

- It is not clear how many independent experiments (not replicates) were performed, which is important for the primary cells.

- Data of controls (antisense siRNA) is not mentioned.

- In which media in addition to distilled water the particles were characterized? If BSA should stabilize particles in physiological solutions, characterization in medium would be relevant.

- Few errors in typing or grammar need correction. Examples: l.44, l.180,

Author Response

Thanks for reviewer's comments and the point-to-point reponses are listed below.

1. Although it is known that the carriers are biocompatible it would be useful to know at which concentration they become cytotoxic in these cells.

Response:  The toxicity information was provided in line 3-4 in Section 4 (Discussion) (page 18).

2. It is not clear how many independent experiments (not replicates) were performed, which is important for the primary cells.

Response:  these ecperiemnts were done triplicate (Figures 2, 3, and 5), as included in the caption.

3. Data of controls (antisense siRNA) is not mentioned.

Response:  We did these control expreriments, which shows that there was no any obvious knockdown of target gene, as mentioned in the revised version (page 14). 

4. In which media in addition to distilled water the particles were characterized? If BSA should stabilize particles in physiological solutions, characterization in medium would be relevant.

Responses: The particles were charcterized in water, PBS and media. Without BSA coating, well dispsersed LDH NPs in pure water will precipitate in PBS and media. With BSA coating, LDH-BSA will keep the dispersion even in PBS and media, and the details were reported in ref 20.

5. Few errors in typing or grammar need correction. Examples: l.44, l.180,

Response:  The authors tried to imrove the languages and did a number of revisions and editings, as highlighted in red font.

Round  2

Reviewer 2 Report

The authors addressed my comments